# Effect of Co, Ti and Cr Additions on Microstructure, Magnetic Properties and Corrosion Resistance of Magnetocaloric Gd-Ge-Si Alloys

**DOI:** 10.3390/ma13245758

**Published:** 2020-12-17

**Authors:** Mariusz Hasiak, Jacek G. Chęcmanowski, Barbara Kucharska, Amadeusz Łaszcz, Aleksandra Kolano-Burian, Jerzy Kaleta

**Affiliations:** 1Department of Mechanics, Materials and Biomedical Engineering, Wrocław University of Science and Technology, 25 Smoluchowskiego, 50-370 Wrocław, Poland; amadeusz.laszcz@pwr.edu.pl (A.Ł.); jerzy.kaleta@pwr.edu.pl (J.K.); 2Department of Advanced Materials Technologies, Wrocław University of Science and Technology, 25 Smoluchowskiego, 50-370 Wrocław, Poland; jacek.checmanowski@pwr.edu.pl; 3Department of Materials Engineering, Częstochowa University of Technology, 19 Armii Krajowej, 42-200 Częstochowa, Poland; barbara.kucharska@pcz.pl; 4Łukasiewicz Research Network, Institute of Non-Ferrous Metals, 5 Sowińskiego, 44-100 Gliwice, Poland; olak@imn.gliwice.pl

**Keywords:** GdGeSi-based alloys, magnetocaloric effect, microstructure, magnetic properties, corrosion resistance

## Abstract

The paper presents studies of microstructure, magnetic and corrosion properties of the Gd_58_Ge_20_Si_22_, Gd_56_Ge_20_Si_22_Co_2_, Gd_56_Ge_20_Si_22_Ti_2_ and Gd_56_Ge_20_Si_22_Cr_2_ (at.%) alloys after isothermal heat treatment at 1450 K for 2 h. The structure investigations of the produced materials performed by X-ray diffraction show the presence of Gd_5_Ge_2_Si_2_-type phase in all investigated samples. DC and AC magnetic measurements confirmed that the Curie temperature depends on the chemical composition of the produced alloys. From *M*(*T*) characteristics, it was found that the lowest Curie point (*T_C_* = 268 K) was estimated for the Gd_58_Ge_20_Si_22_ sample, whereas the highest value of the Curie temperature (*T_C_* = 308 K) was for the Gd_56_Ge_20_Si_22_Cr_2_ alloys. Moreover, the GdGeSi alloy without alloying additions shows the highest magnetic entropy change |ΔS_M_| = 15.07 J⋅kg^−1^⋅K^−1^ for the maximum magnetic field of 2 T. The maximum |ΔS_M_| measured for the Gd_56_Ge_20_Si_22_ with the addition of Co, Ti or Cr for the same magnetic field was obtained in the vicinity of the Curie point and equals to 2.92, 2.73 and 2.95 J⋅kg^−1^⋅K^−1^, respectively. Electrochemical studies of the produced materials for 60 min and 55 days exposure in 3% NaCl solution show that the highest stability and corrosion resistance were exhibited the sample with added of Ti.

## 1. Introduction

Rare earth (RE) based alloys are very attractive materials because of their good magnetic properties [1,2,3,4]. It has been reported in several papers that materials with the addition of rare earth elements e.g., Gd [5,6], Pr [6,7,8,9], Nd [6,10,11], Sm [8], Tb [5] etc. are widely used in many devices, mostly electrical, due to their distinctive thermomagnetic characteristics. When it comes to potential applications in magnetic devices, some of the most interesting materials are Gd-based alloys [12,13,14,15,16,17,18]. These alloys are intensively investigated because they exhibit high magnetocaloric effect (MCE), represented by magnetic entropy changed (|ΔS_M_|), in the vicinity of the Curie point, which is close to the room temperature [18,19,20]. It was also shown that the Curie temperature could be changed by the introduction to master alloys additives such as Co, Fe, Zr, Pr, Ni or Ce [9,16,18,21,22]. High MCE in these alloys allows them to be used in cooling technology, for example, in the production of ecofriendly magnetic refrigerators [23,24,25,26,27]. The possibilities of increasing magnetocaloric effect in RE-based materials by proper heat treatment and/or chemical composition changes, together with reducing hysteresis loss [28,29,30,31] and improving application properties, such as corrosion resistance and stability [15,32,33,34] as well as mechanical properties [35,36,37], is one of the most important issues in research on these alloys.

The aim of this paper is to study the effect of Co, Ti and Cr addition on microstructure, DC and AC thermomagnetic properties as well as corrosion stability and resistivity for the Gd_58-x_Ge_20_Si_22_Y_x_ (x = 0 or 2; Y = Co, Ti or Cr) alloys after the heat treatment at 1450 K for 2 h. Particular attention in this work is devoted to the influence of additives on the Curie temperature change, which is essential from the application point of view. The introduction of Co, Ti and Cr atoms to master GdGeSi alloy, besides influence on microstructure and thermomagnetic properties, changes electrochemical properties. The main reason for using the alloying additives stems from the fact that in the presence of Cr, the oxidation resistance of metallic materials increases, which leads to the creation of a dense oxide layer and limits the rate of corrosion processes. A similar effect is also observed in Co- and Ti-containing alloys.

## 2. Materials and Methods

GdGeSi-based alloys with additions of Co, Ti or Cr and a nominal compositions of Gd_58_Ge_20_Si_22_, Gd_56_Ge_20_Si_22_Co_2_, Gd_56_Ge_20_Si_22_Ti_2_ and Gd_56_Ge_20_Si_22_Cr_2_ (at.%) were produced with the help of an arc melter system in an argon atmosphere using high-purity elements. The total mass of the produced samples was about 10 g. The ingots were remelted five times in order to achieve a homogeneous structure. All samples were subjected to isothermal annealing at 1450 K for 2 h. The temperature for the heat treatment process was chosen according to differential scanning calorimetry (DSC) measurements. The phase composition of the Gd_58-x_Ge_20_Si_22_Y_x_ alloys, where x = 0 or 2 and Y = Co, Ti or Cr, was examined by X-ray diffraction (XRD) with the help of a Seifert 3003TT diffractometer (Seifert, Mannheim, Germany) working in Bragga–Brentano geometry for CoK_α_ = 0.17902 nm radiation in the angle range 2Θ = 35–55° for the samples in the form of powders. The structure studies and chemical composition analysis of the produced materials were performed by both optical microscopy and scanning electron microscopy equipped with an energy dispersive X-ray spectroscopy (SEM, Quanta 250, FEI, Thermo Fisher Scientific, Waltham, MA, USA) detector working in secondary electrons (SE) mode. Thermomagnetic measurements, i.e., magnetization and magnetic susceptibility versus temperature for the Gd_58_Ge_20_Si_22_ as well as Co-, Ti- and Cr-containing samples, were carried out by the Versalab and Physical Property Measurement System (PPMS, Quantum Design, San Diego, CA, USA) to determine the Curie point (*M*(*T*) characteristics recorded at the external magnetic field of 0.25 T) and magnetocaloric effect (MCE) described as magnetic entropy changes (ΔS_M_). DC magnetic measurements (temperature dependence of magnetization, sets of isothermal magnetization curves) were performed at temperature range 50–400 K and extremal magnetic field up to 2 T. AC magnetic susceptibility (ACMS) was recorded from 50 to 350 K for excitation magnetic field of 0.5 mT and frequency of 100 Hz at zero external DC magnetic field. Corrosion resistance tests were conducted in 3% NaCl solution for 60 min and 55 days. DC electrochemical measurements were conducted by recording polarization curves in the conventional three-electrode system. The measuring setup included a measuring vessel and Schlumberger SI 1286 potentiostat (Ametek, Berwyn, PA, USA). The measurements started from the potential of –350 mV in the anode direction with a rate of 1 mV/s.

## 3. Results and Discussion

### 3.1. Structure and Phase Investigations of the Gd_58-x_Ge_20_Si_22_Y_x_ (x = 0 or 2; Y = Co, Ti or Cr) Alloys

The investigation of microstructure performed with the help of X-ray diffractometer for the Gd_58-x_Ge_20_Si_22_Y_x_ (x = 0 or 2; Y = Co, Ti or Cr) alloys after isothermal annealing at 1450 K for 2 h is presented in Figure 1. It can be seen that diffraction pattern recorded for diffraction angle 2Θ = 35–55° for all investigated materials shows the presence of peaks, which can be assigned to the required high magnetocaloric Gd_5_Ge_2_Si_2_-type phase. The changes in the intensity of peaks between investigated samples reflect additions of Co, Ti and Cr to master Gd_58_Ge_20_Si_22_ alloy and texture of the examined materials. It is particularly evident for the reflex described by d-spacing in Bragg’s law equals to d_hkl_ = 2.68 Å for the (0 4 2) plane (PDF cards: 04-008-5158 or 01-073-3500).

The microstructure and phase composition studies of the produced materials performed by an optical microscope and an SEM in SE mode equipped with an EDS detector are presented in Figure 2. The micrographs of the Gd_58_Ge_20_Si_22_, Gd_56_Ge_20_Si_22_Co_2_, Gd_56_Ge_20_Si_22_Ti_2_ and Gd_56_Ge_20_Si_22_Cr_2_ recorded with the help of OM (Figure 2, left column) show the presence of dendrites with different sizes precipitated during the annealing process. The images of fracture of the Gd_58-x_Ge_20_Si_22_Y_x_ (x = 0 or 2; Y = Co, Ti or Cr) alloys obtained from SEM/SE presented in the middle column in Figure 2 shows no significant changes between investigated samples and suggests similar brittleness of the produced materials despite the noticeable difference in the microstructure. The investigations of chemical composition for all samples performed by SEM with EDS analysis were conducted to confirm the nominal 5:2:2 ratio for the Gd:Ge:Si phase. The obtained results are in good agreement with X-ray examination data. Moreover, the presence of alloying additions such as Co, Ti and Cr in GdGeSi-based alloy is evident as additional peaks on the EDS spectra (Figure 2, right column).

### 3.2. Magnetic Properties of GdGeSi Alloy without and with the Addition of Co, Ti and Cr

Figure 3 (left side) shows the temperature dependence of magnetization (*M*(*T*)) for the annealed Gd_58-x_Ge_20_Si_22_Y_x_ (x = 0 or 2; Y = Co, Ti or Cr) alloys performed with a heating rate of 10 K/min in zero field-cooled mode at the external magnetic field of 0.25 T recorded in the temperature range 50–400 K.

The Curie point (*T_C_*) for all analyzed samples calculated as a temperature, which corresponds to the minimum value of the derivative *dM*/*dT*, is presented in Figure 3 (right side). The recorded *M*(*T*) thermomagnetic characteristics are typical for ferromagnetic materials with first (FOPT) and second (SOPT) order phase transition. The Gd_58_Ge_20_Si_22_ sample shows the lower Curie point (*T_C_* = 268 K) in comparison to sample with the addition of Co, Ti and Cr (304 K, 298 K and 308 K, respectively). The addition 2 at.% of Co, Ti or Cr to master GdGeSi alloy leads to an increase of the Curie point, which is important from the application point of view. On the other hand, the magnetic magnetization measured at the same temperature in the ferromagnetic region (50–250 K) varies with the chemical composition of produced materials. The Gd_58_Ge_20_Si_22_ sample exhibits higher magnetization than samples with additives (Figure 3). Moreover, the reference GdGeSi sample shows FOTP [16], whereas Co-, Ti and Cr-containing materials present ferromagnetic to paramagnetic phase transition described as SOPT. It is worth noting that the smallest value of *M* in the temperature range 50–250 K for all investigated samples was observed for the Co-containing sample despite the highest magnetic moment.

The estimation of the Curie temperature for the annealed GdGeSi-based alloys with the addition of Co, Ti and Cr was also performed by registering AC magnetic susceptibility versus temperature for the excitation magnetic field of 0.5 mT, without an external magnetic field (Figure 4). It is clear that the values of the Curie point are in good agreement with the results obtained from *M*(*T*) analysis and the Curie temperature for the Gd_58_Ge_20_Si_22_, Gd_56_Ge_20_Si_22_Co_2_, Gd_56_Ge_20_Si_22_Ti_2_ and Gd_56_Ge_20_Si_22_Cr_2_ alloys equals to 267 K, 304 K, 297 K and 307 K, respectively. The difference between the Curie points obtained from magnetization and ACMS versus temperature measurements are due to the external magnetic field, which was applied only during *M*(*T*) investigations. It is also worth noting that on the AC magnetic susceptibility versus temperature curves just below the Curie point the Hopkinson peak for Ti- and Cr-containing sample was observed. This effect seems to be due to the incoherent rotation of the magnetization during the reversal process when AC excitation magnetic field is applied [38].

Figure 5 presents the set of isothermal *M*(*μ*_0_*H*) curves (left column) and corresponding magnetocaloric effect investigations (right column) for the annealed Gd_58-x_Ge_20_Si_22_Y_x_ (x = 0 or 2; Y = Co, Ti or Cr). MCE is described as isothermal magnetic entropy change (Δ*S_M_*) and can be calculated according to Maxwell’s thermomagnetic relation [39]:(1)$$\Delta S_{M}=\mu_{0}\int_{0}^{H_{m}}{\left(\frac{\partial M\left(T,H\right)}{\partial T}\right)}_{H}dH=\int_{0}^{B_{m}}\frac{\partial M\left(T,H\right)}{\partial T}dB$$
where *B_m_* = *μ*_0_*H_m_*. The changes of magnetic entropy (Δ*S_M_*) versus temperature were calculated from the sets of isothermal magnetization curves (Figure 5, left column), recorded for maximum magnetizing field up to 2 T, using numerical approximation described by the equation:(2)$$\left|\Delta S_{M}\right|=\sum_{i}\frac{1}{T_{i+1}-T_{i}}{\left(M_{i}-M_{i+1}\right)}_{H}\Delta H_{i}$$
where *M_i_* and *M_i_*_+1_ are magnetizations measured for magnetizing field *H* at temperatures *T_i_* and *T_i_*_+1_, respectively.

*M*(*μ*_0_*H*) characteristics recorded in the vicinity of the Curie point (Figure 5 left column) confirm the FOPT in the Gd_58_Ge_20_Si_22_ and SOPT in the Gd_56_Ge_20_Si_22_Y_2_ (Y = Co, Ti or Cr). The increase of magnetizing field from 0.5 T to 2 T leads to increase of the maximum value of |Δ*S_M_*| from 4.33 to 15.07 J⋅kg^−1^⋅K^−1^, from 0.91 to 2.92 J⋅kg^−1^⋅K^−1^, from 0.92 to 2.73 J⋅kg^−1^⋅K^−1^ and from 0.96 to 2.95 J⋅kg^−1^⋅K^−1^ for the Gd_58_Ge_20_Si_22_, Gd_56_Ge_20_Si_22_Co_2_, Gd_56_Ge_20_Si_22_Ti_2_ and Gd_56_Ge_20_Si_22_Cr_2_ alloys, respectively. Moreover, the maximum on −Δ*S_M_*(*T*) characteristics was observed close to the Curie point of the investigated materials. Obtained results revealed in Figure 5 suggest that the addition of Co, Ti or Cr atoms to GdGeSi-based alloy drastically reduce |Δ*S_M_*|. Moreover, Co-, Ti- and Cr-containing samples present near to Curie point almost the same maximum values of magnetocaloric effect, which are more than five times lower than for the Gd_58_Ge_20_Si_22_ alloy. The similar thermomagnetic behaviour was also reported for the magnetocaloric Gd-based alloys doped by a different alloying element such as Zr, Mn, Tb, Fe [9,12,16,18,22].

### 3.3. Corrosion Properties of GdGeSi-Based Alloys

Due to the good corrosion resistance of Co, Ti and Cr, the influence of these additives on the corrosion behavior of the Gd_58-x_Ge_20_Si_22_Y_x_ (x = 0 or 2; Y = Co, Ti or Cr) alloys was investigated. From an application point of view, the corrosion resistance of the investigated magnetocaloric materials is significant because these alloys usually work in chloride-containing water used as a cooling medium in environmentally friendly refrigeration. All investigations of electrochemical properties presented below were performed for the Co-, Ti- and Cr-containing samples in relation to the Gd_58_Ge_20_Si_22_ as the reference sample. Figure 6 and Figure 7 present the corrosion properties of the investigated alloys after 60 min. and 55 days exposition in 3% NaCl solution, respectively. Figure 6 shows that for the Gd_58_Ge_20_Si_22_ sample and with the addition of Co, Ti and Cr, even for a short exposure time (60 min. in 3% NaCl solution), differences in the rates of the electrode processes proceeding on the metallic surface were registered.

The electrode processes proceed the slowest, in both the anode region and the cathode region, on the surface of Ti-containing sample. The presence of Co and Cr in the GdGeSi alloys causes the electrode processes to proceed more easily than in the case of the metallic substrate without the alloy additions. This is due to the difference in passivation presented by Co, Ti and Cr. Among the used alloying elements, Ti shows the strongest tendency towards surface passivation. The presence of Co, Ti and Cr in the matrix of the GdGeSi alloy does not show a significant effect on the values of the particular electrochemical parameters (Table 1). After 60 min exposure in the corrosive solution, the maximum differences between the corrosion potential (*E_corr_*) values of 44 mV and the cathodic–anodic transition potential (*E_C-A_*) values of 28 mV were measured for the Gd_56_Ge_20_Si_22_Cr_2_ and Gd_58_Ge_20_Si_22_ alloys (Table 1). The highest polarization resistance (*R_p_* = 4.18 × 10^6^ Ω⋅cm^2^) and the lowest rate of corrosion (*i_corr_* = 6.23 × 10^−9^ A/cm^2^) were recorded for the Gd_56_Ge_20_Si_22_Ti_2_ alloy. The rate of corrosion for this material is about 15 times lower than for the Cr-containing material. Moreover, the Gd_56_Ge_20_Si_22_Ti_2_ and Gd_58_Ge_20_Si_22_ alloys present comparable corrosion resistance. On the basis of the obtained data, it can be concluded that the Gd_56_Ge_20_Si_22_Ti_2_ alloy after 60 min exposure in 3% NaCl solution presents the highest corrosion resistance, whereas the Gd_56_Ge_20_Si_22_Cr_2_ alloy exhibits the lowest corrosion resistance.

The long-time (55 days) exposure in 3% NaCl solution for the Gd_58-x_Ge_20_Si_22_Y_x_ (x = 0 or 2; Y = Co, Ti or Cr) alloys leads to the significant differences in the rates of the electrode processes in both the cathode and anode region in comparison to 60 min exposure in the same solution (Figure 7, Table 1). Taking into account the obtained values of polarization resistance and rates of corrosion for all investigated samples (55 days exposure time), the Gd_58_Ge_20_Si_22_ and Gd_56_Ge_20_Si_22_Ti_2_ alloys show the highest (in comparison to samples with Co and Cr) and comparable corrosion resistance. The resulting passive layer of titanium oxide effectively limits the rate of electrode processes in the anode area and does not tend to allow pitting corrosion (as in the case of an alloy containing Co or Cr).

For the Gd_58−x_Ge_20_Si_22_Y_x_ (x = 0 or 2; Y = Co, Ti or Cr) alloys the best stability and corrosion resistance after 55 days exposition in the 3% NaCl solution are presented by the Ti-containing alloy. This material shows the highest values of *E_corr_* = −487 mV and *E_C–A_* = −571 mV in comparison to other investigated alloys. Moreover, the polarization resistance (*R_p_*) and current corrosion density (*i_corr_*) for Ti-containing sample are comparable with data measured for the Gd_58_Ge_20_Si_22_ alloys (Table 1). It is also clear that the Gd_56_Ge_20_Si_22_Co_2_ and Gd_56_Ge_20_Si_22_Cr_2_ alloys show similar corrosion properties; however, they are much worse than the Gd_56_Ge_20_Si_22_Ti_2_ and Gd_58_Ge_20_Si_22_ materials. From electrochemical investigations presented above (Figure 6 and Figure 7, Table 1) it can be concluded that relatively the highest stability and corrosion resistance in 3% NaCl solution are shown by the sample with the addition of Ti.

## 4. Conclusions

The paper presents the influence of Co, Ti and Cr additions to GdGeSi-based alloy on the Curie temperature change, magnetocaloric effect as well as corrosion stability for the Gd_58-x_Ge_20_Si_22_Y_x_ (x = 0 or 2; Y = Co, Ti or Cr) alloys. X-ray diffraction shows the presence of master Gd_5_Ge_2_Si_2_-type phase in all investigated samples. The additional SEM/EDS analysis confirms the presence of Co, Ti and Cr in GdGeSi-based alloy as well as the nominal 5:2:2 ratio for the Gd:Ge:Si phase. From an application point of view all additives studied in this paper increase the Curie point from *T_C_* = 268 K for the Gd_58_Ge_20_Si_22_ alloy to temperatures which are close to room temperature, i.e., 304 K, 298 K and 308 K for the Co-, Ti and Cr-containing alloys, respectively. The magnetocaloric effect, described as the maximum magnetic entropy change, observed near the Curie point strongly depends on chemical composition and, for the Gd_58_Ge_20_Si_22_ sample (|ΔS_M_| = 15.07 J⋅kg^−1^⋅K^−1^), is more than five times higher than for the materials with 2 at.% addition of Co, Ti and Cr (2.92, 2.73 and 2.95 J⋅kg^−1^⋅K^−1^, respectively). On the other hand, it should be noted that this lower MCE in doped alloys occurs in the vicinity of the room temperature and is still valuable from an application point of view. Moreover, the electrochemical data shows an improvement of corrosion resistance for all doped materials in comparison to the main Gd_58_Ge_20_Si_22_ alloy. It is worth noting that the best corrosion resistance in 3% NaCl solution was observed for the Ti-containing sample. The same Gd_56_Ge_20_Si_22_Ti_2_ alloy exhibits the strongest tendency towards surface passivation during both 60 min and 55 days of exposure (polarization resistance of 4.18 × 10^6^ Ω⋅cm^2^ and 3.62 × 10^6^ Ω⋅cm^2^ as well as corrosion current density of 6.26 × 10^−9^ A/cm^2^ and 7.21 × 10^−9^ A/cm^2^, respectively).

## Figures and Tables

**Figure 1 materials-13-05758-f001:**
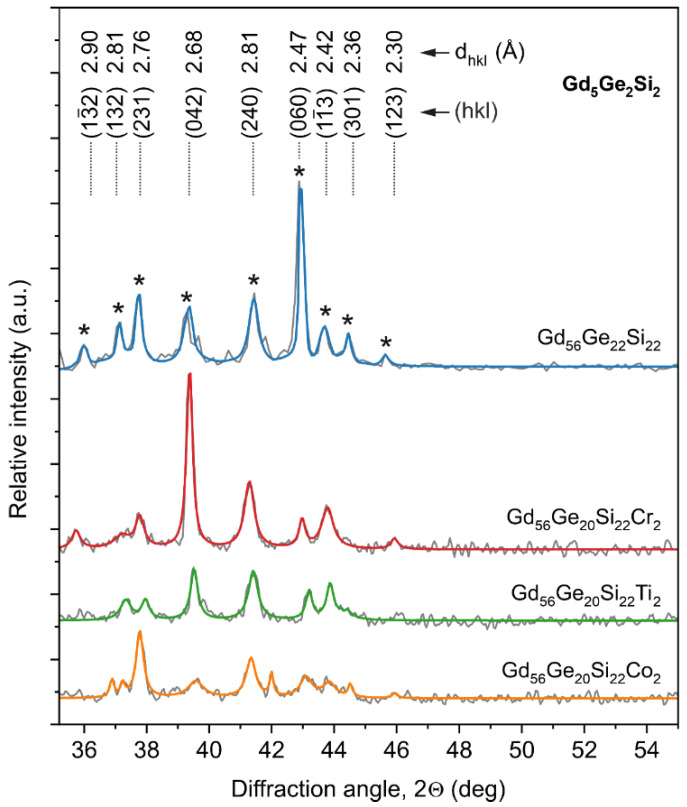
X-ray diffraction pattern of the Gd_58-x_Ge_20_Si_22_Y_x_ (x = 0 or 2; Y = Co, Ti or Cr) alloys after heat treatment at 1450 K for 2 h. The peaks corresponding to Gd_5_Ge_2_Si_2_-type phase are marked with an asterisk.

**Figure 2 materials-13-05758-f002:**
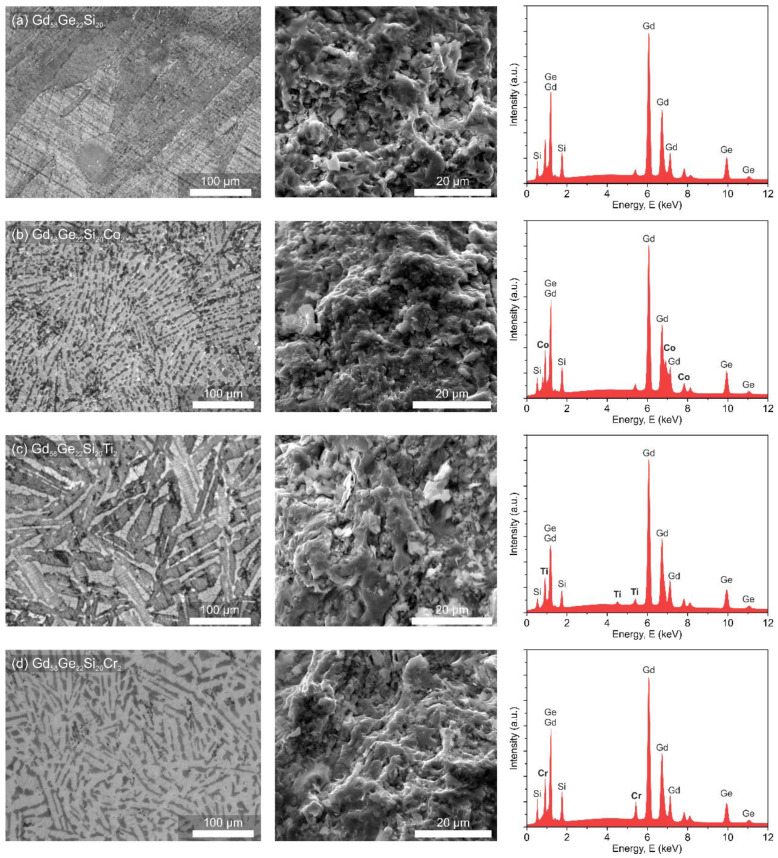
Optical microscopy images (left column), SEM/SE fracture images (middle column) and corresponding EDS spectra (right column) for the annealed Gd_58_Ge_20_Si_22_ (**a**), Gd_56_Ge_20_Si_22_Co_2_ (**b**), Gd_56_Ge_20_Si_22_Ti_2_ (**c**) and Gd_56_Ge_20_Si_22_Cr_2_ (**d**) alloys.

**Figure 3 materials-13-05758-f003:**
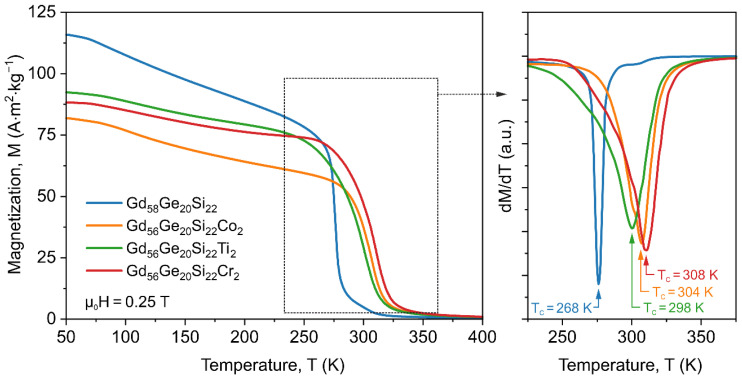
Magnetic magnetization versus temperature curves for the Gd_58_Ge_20_Si_22_, Gd_56_Ge_20_Si_22_Co_2_, Gd_56_Ge_20_Si_22_Ti_2_ and Gd_56_Ge_20_Si_22_Cr_2_ alloys annealed at 1450 K for 2 h recorded at external DC magnetic field of 0.25 T.

**Figure 4 materials-13-05758-f004:**
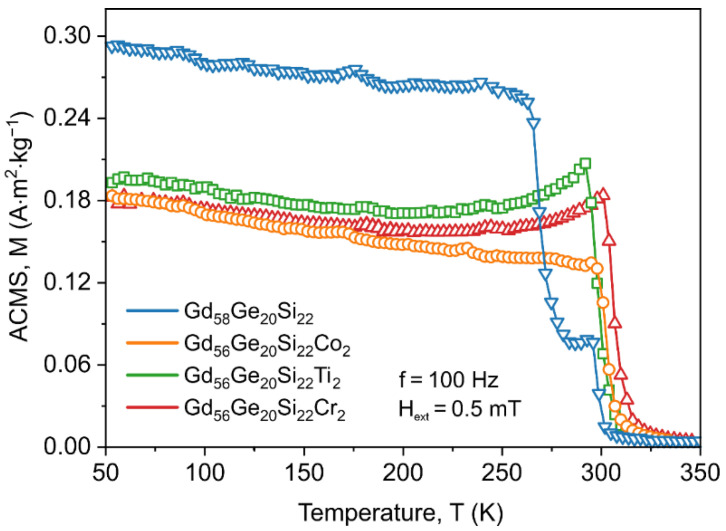
Temperature dependence of AC magnetic susceptibility (ACMS) for the annealed Gd_58_Ge_20_Si_22_, Gd_56_Ge_20_Si_22_Co_2_, Gd_56_Ge_20_Si_22_Ti_2_ and Gd_56_Ge_20_Si_22_Cr_2_ alloys measured at zero external magnetic field for the excitation AC magnetic field of 0.5 mT and frequency of 100 Hz.

**Figure 5 materials-13-05758-f005:**
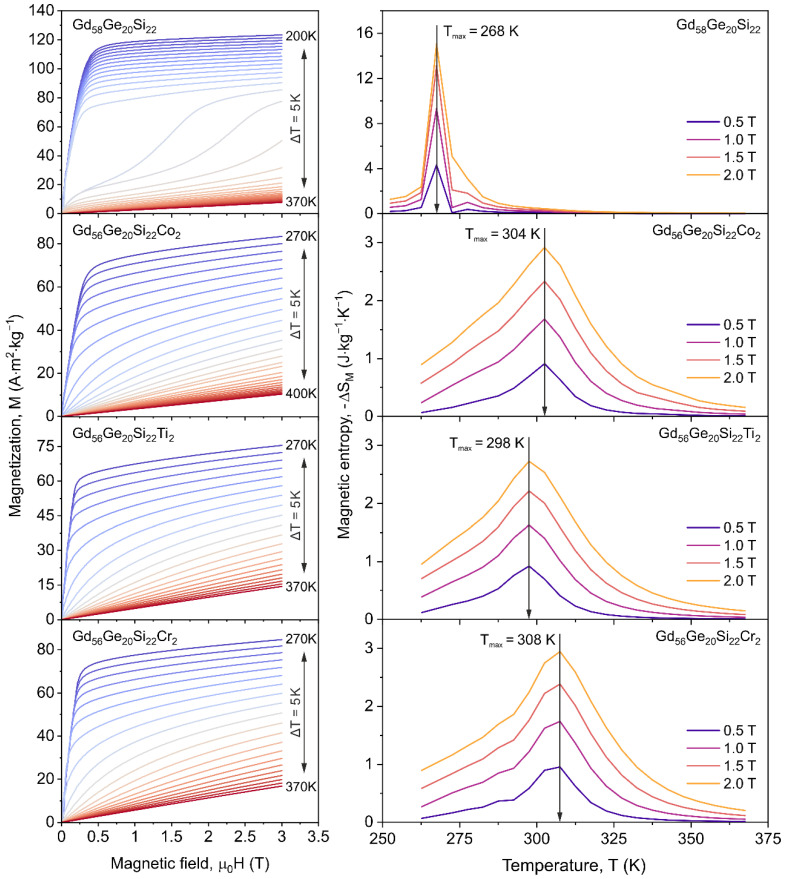
Set of isothermal M(*μ*_0_*H*) curves and corresponding magnetic entropy changes versus temperature calculated for the Gd_58-x_Ge_20_Si_22_Y_x_ (x = 0 or 2; Y = Co, Ti or Cr) alloys at the maximum external magnetizing field of 0.5 T, 1.0 T, 1.5 T and 2.0 T.

**Figure 6 materials-13-05758-f006:**
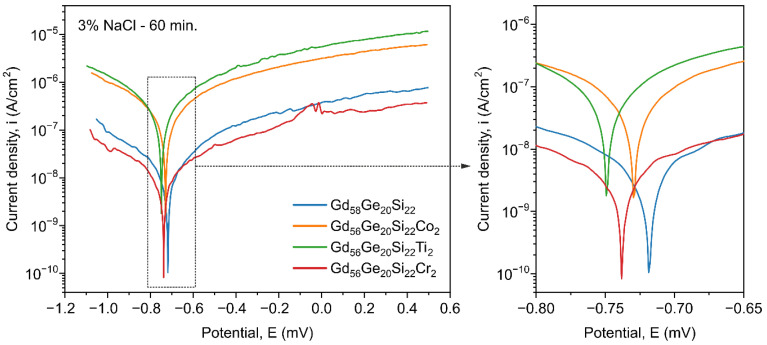
Polarisation curves determined in 3% NaCl solution for the Gd_58-x_Ge_20_Si_22_Y_x_ (x = 0 or 2; Y = Co, Ti or Cr) alloys after 60 min exposure.

**Figure 7 materials-13-05758-f007:**
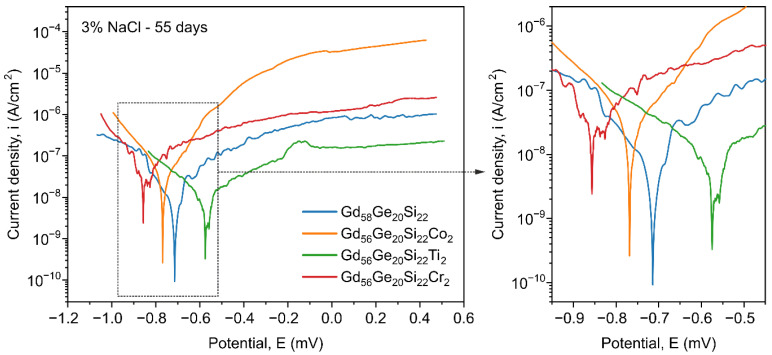
Polarisation curves determined in 3% NaCl solution for the the Gd_58-x_Ge_20_Si_22_Y_x_ (x = 0 or 2; Y = Co, Ti or Cr) alloys after 55 days exposure.

**Table 1 materials-13-05758-t001:** Corrosion potential (*E_corr_*), cathodic to anodic transition potential (*E_C–A_*), polarization resistance (*R_p_*) and corrosion current density (*i_corr_*) for the Gd_58_Ge_20_Si_22_, Gd_56_Ge_20_Si_22_Co_2_, Gd_56_Ge_20_Si_22_Ti_2_ and Gd_56_Ge_20_Si_22_Cr_2_ alloys in 3% NaCl solutions after two exposure times.

Time of Exposition	Alloy	*E_corr_* (mV)	*E_C-A_* (mV)	*R_p_* (Ω∙cm^2^)	*i_corr_* (A/cm^2^)
60 min	Gd_58_Ge_20_Si_22_	−707	−720	3.46 × 10^6^	7.54 × 10^−9^
	Gd_56_Ge_20_Si_22_Co_2_	−729	−729	2.90 × 10^5^	8.98 × 10^−8^
	Gd_56_Ge_20_Si_22_Ti_2_	−737	−737	4.18 × 10^6^	6.26 × 10^−9^
	Gd_56_Ge_20_Si_22_Cr_2_	−751	−748	2.27 × 10^5^	1.15 × 10^−7^
55 days	Gd_58_Ge_20_Si_22_	−721	−713	3.56 × 10^6^	7.33 × 10^−9^
	Gd_56_Ge_20_Si_22_Co_2_	−647	−766	7.49 × 10^5^	3.48 × 10^−8^
	Gd_56_Ge_20_Si_22_Ti_2_	−487	−571	3.62 × 10^6^	7.21 × 10^−9^
	Gd_56_Ge_20_Si_22_Cr_2_	−704	−855	5.06 × 10^5^	5.15 × 10^−8^

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
