# Peer review of "Effect of Co, Ti and Cr Additions on Microstructure, Magnetic Properties and Corrosion Resistance of Magnetocaloric Gd-Ge-Si Alloys"

_materials, 2020, doi:10.3390/ma13245758_

Round 1
Reviewer 1 Report
In this manuscript the authors claim to study four alloys: pure Gd-Ge-Si and Cr, Co and Ti doped alloys to compare between them. The paper contains too many unclear points and I propose to reject it. The English need a substantial improvement.
My comments are as follows:
My main concern is the lack of uncertainty values in all data provide in the text and in Table 1. E.g. the Tc values obtained from dc and ac magnetic studies may be within the uncertainties the same.
Page 2 line 64. What are the annealing conditions? ambient atmosphere? inert gas atmosphere? What was the cooling rate ? etc?
Page 2 line 74. What is the city name ?
Fig. 1. What are the Miller indices, the lattice parameters and the crystal structure? There are some minor peaks which are not addressed.
Page 3 line 101. It should be Fig. 2 not Fig. 1.
Fig. 2 right. The peaks related to the dopants are higher than their concentration 2%. The authors should list the exact concentration of each element deduced from the EDS data.
Fig. 3. Indeed, the transition for the undoped material is sharper but the claim for FOPT is not proven. Moreover Fig. 4 shows two transitions which are not explained.
Page 5 lines 130-131. This claim needs isothermal M(H) plots, in order to show definitely the reduced moments of the doped materials.
Fig.5. The high fields experimental data from which fig. 5 is constructed must be shown.
Doping should increase the entropy, in contrast to Fig. 5
Author Response
Dear Reviewer,
thank you very much for the revision of our paper and your suggestions. Below you can find our response marked red.
Sincerely yours,
Mariusz Hasiak
-------------------------------------------------------------------------------
In this manuscript the authors claim to study four alloys: pure Gd-Ge-Si and Cr, Co and Ti doped alloys to compare between them. The paper contains too many unclear points and I propose to reject it. The English need a substantial improvement.
English was carefully checked and corrected.
My comments are as follows:
My main concern is the lack of uncertainty values in all data provide in the text and in Table 1. E.g. the Tc values obtained from dc and ac magnetic studies may be within the uncertainties the same.
All DC electrochemical measurements were conducted by a measuring vessel and Schlumberger SI 1286 potentiostat under the same condition. Moreover, the results presented in the manuscript and Table 1 were measured with the same uncertainties. Therefore, the measuring error of data in Table 1 does not influence on direct comparison of recorded parameters between investigated samples.
The discontinuous transformation from ferro to paramagnetic state undergoing in the vicinity of Curie point (TC) is described as SOPT and TC is usually estimated as an inflection point on M(T) curve. According to this there is no importance to calculate uncertainties of TC.
Page 2 line 64. What are the annealing conditions? ambient atmosphere? inert gas atmosphere? What was the cooling rate ? etc?
The samples were sealed in a vacuumed quartz tubes. After annealing all materials were cooled down to room temperature in the furnace chamber.
Page 2 line 74. What is the city name ?
The city name (Quantum Design, San Diego, CA, USA) was added to the text.
Fig. 1. What are the Miller indices, the lattice parameters and the crystal structure? There are some minor peaks which are not addressed.
The Miller indices were added to Fig. 1. This figure shows only peaks, which correspond to the main Gd5Ge2Si2 phase detected in all investigated materials.
Page 3 line 101. It should be Fig. 2 not Fig. 1.
It was changed according to the suggestion.
Fig. 2 right. The peaks related to the dopants are higher than their concentration 2%. The authors should list the exact concentration of each element deduced from the EDS data.
The chemical concentration of investigated alloys was estimated by using EDAX TEAM SMART INSIGHT software calibrated for EDS detector (SEM, Quanta 250, FEI). The deconvolution process of the whole EDS spectrum allows determining the chemical composition of materials and ratio of each alloying element in total concentration. It means that it is not possible to determine the amount of every single element based on only one peak on EDS spectrum.
Fig. 3. Indeed, the transition for the undoped material is sharper but the claim for FOPT is not proven. Moreover Fig. 4 shows two transitions which are not explained.
FOPT in GdGeSi alloys was also studied by the temperature dependence of heat capacity - described in detail in ref. [16]. On the base of these results authors have written this sentence following ref. [16].
Page 5 lines 130-131. This claim needs isothermal M(H) plots, in order to show definitely the reduced moments of the doped materials.
Isothermal M(m0H) curves for all investigated alloys were added to Fig. 5 (left column). The reference to this figure (Fig. 5, left column) was added in the text (lines 130-131).
Fig.5. The high fields experimental data from which fig. 5 is constructed must be shown.
According to the suggestion of the reviewer, the set of experimental high fields isothermal M(m0H) curves for all investigated materials were added to Fig. 5, left column.
Doping should increase the entropy, in contrast to Fig. 5
According to our studies the additives such as Co, Ti and Cr change the Curie temperatures and reduced magnetic entropy changes in investigated alloys.
Reviewer 2 Report
This work presents the studies on the effect of Co, Ti and Cr additions on microstructure, magnetic properties and corrosion resistance of magnetocaloric Gd-Ge-Si alloys The authors discuss the structure of the produced materials and the presence of Gd5Ge2Si2-type phase in all investigated samples. They evaluate its influence on the Curie temperature of the produced alloys.
The electrochemical studies of the produced materials are also present. The highest stability and corrosion resistance of the sample with addition of Ti was shown. These effects were investigated deeply and the conclusions are supported by the experimental data.
I recommend this paper to be published in the present form.
Author Response
Dear Reviewer,
thank you very much for the revision of our paper.
Sincerely yours,
Mariusz Hasiak
Reviewer 3 Report
Paper is devoted to the studies of the magnetic, magnetocaloric and corrosion effects in Gd-Ge-Si samples with Co, Cr, Ti doping.
The significant reduction of magnetocaloric properties in doped Gd-Ge-Si-Y samples were observed that is weak point of this paper.
Reviewer have several notes, which need in explanation from authors:
- What was the main idea of doping of Gd-Ge-Si alloy? If it was in enhacnement of MCE- results are not so good..effect decreases about 5 times. If the idea was in improovement of anti corrosion properties - please add more information in introduction part. Why corrosian stability is important? please, motivate this point in introduction
- The results observed on Fig 5 are explained briefly Line 166-173. There are no analysis and comparasion with literature data
- Conclusions looks as counting of the experimental results. I think that in this part should be presented main summarized result of this work in connection with aim . Conclusion should be rewrited.
Author Response
Dear Reviewer,
thank you very much for the revision of our paper and your suggestions. Below you can find our response marked red.
Sincerely yours,
Mariusz Hasiak
-------------------------------------------------------------------------------
Paper is devoted to the studies of the magnetic, magnetocaloric and corrosion effects in Gd-Ge-Si samples with Co, Cr, Ti doping.
The significant reduction of magnetocaloric properties in doped Gd-Ge-Si-Y samples were observed that is weak point of this paper.
The main reason for this studies was to investigate the influence of Co, Ti and Cr additions to GdGeSi-based alloy on the Curie temperature change. The Curie point close to room temperature is significant from an application point of view. The following sentence “This paper aims to study the effect of Co, Ti and Cr addition on microstructure, DC and AC thermomagnetic properties as well as corrosion stability and resistivity for the Gd58 xGe20Si22Yx (x = 0 or 2; Y = Co, Ti or Cr) alloys after the heat treatment at 1450 K for 2 h. Particular attention in this work is devoted to the influence of additives on the Curie temperature change, which is important from the application point of view. The introduction of Co, Ti and Cr atoms to master GdGeSi alloy besides influence on microstructure and thermomagnetic properties also changes electrochemical properties.” was added to the introduction.
Reviewer have several notes, which need in explanation from authors:
- What was the main idea of doping of Gd-Ge-Si alloy? If it was in enhacnement of MCE- results are not so good..effect decreases about 5 times. If the idea was in improovement of anti corrosion properties - please add more information in introduction part. Why corrosian stability is important? please, motivate this point in introduction
The main idea of doping of Gd-Ge-Si alloy was shifting the Curie point close to the room temperature. The information about the influence of Co, Ti and Cr additives on corrosion process are included at the end of the Introduction part. Moreover, the corrosion stability is important because these materials are perspective to be used in ecofriendly magnetic refrigerators with the water-based cooling medium. This explanation is also at the beginning of chapter 3.3.
- The results observed on Fig 5 are explained briefly Line 166-173. There are no analysis and comparasion with literature data
The additional M(m0H) characteristics for all samples were added to Fig. 5 (left column). Moreover, the explanation of results presented in Fig. 5 as well as references for magnetocaloric Gd-based alloys dopped by a different alloying element such as Zr, Mn, Tb, Fe was added to the manuscript.
- Conclusions looks as counting of the experimental results. I think that in this part should be presented main summarized result of this work in connection with aim . Conclusion should be rewrited.
The conclusion was rewritten according to the reviewer’s suggestion.
Round 2
Reviewer 1 Report
The authors addressed all my demands.
Still uncertainties values are very important